# Training Dynamics of Parametric and In-Context Knowledge Utilization in Language Models

## Abstract

Large language models often encounter conflicts between in-context knowledge retrieved at inference time and parametric knowledge acquired during pretraining. Models that accept external knowledge uncritically are vulnerable to misinformation, whereas models that adhere rigidly to parametric knowledge fail to benefit from retrieval. Despite the widespread adoption of retrieval-augmented generation, we still lack a systematic understanding of what shapes knowledge-arbitration strategies during training. This gap risks producing pretrained models with undesirable arbitration behaviors and, consequently, wasting substantial computational resources after the pretraining budget has already been spent. To address this problem, we present the first controlled study of how training conditions influence models' use of in-context and parametric knowledge, and how they arbitrate between them. We train transformer-based language models on a synthetic biographies corpus while systematically controlling various conditions. Our experiments reveal that intra-document repetition of facts fosters the development of both parametric and in-context capabilities. Moreover, training on a corpus that contains inconsistent information or distributional skew encourages models to develop robust strategies for leveraging parametric and in-context knowledge. Rather than viewing these non-ideal properties as artifacts to remove, our results indicate that they are important for learning robust arbitration. These insights offer concrete, empirical guidance for pretraining models that harmoniously integrate parametric and in-context knowledge.

## 1 Introduction

Large language models (Touvron et al., 2023; Brown et al., 2020; Biderman et al., 2023) store and use *parametric knowledge* (Geva et al., 2020; 2023; Meng et al., 2022) acquired during pretraining and increasingly leverage *in-context knowledge* through retrieval-augmented generation (Lewis et al., 2021; Ram et al., 2023; Shi et al., 2023), which supplies external documents at inference time. This allows models to incorporate up-to-date and domain-specific information beyond their training data. A central challenge appears when external documents conflict with parametric knowledge (Neeman et al., 2022), which forces the model to arbitrate between the two sources. The stakes are high when the retrieved content contains misinformation, noisy passages, or adversarially crafted text. Models that trust external sources uncritically become vulnerable to these risks, while models that rigidly rely on their parametric knowledge fail to benefit from valuable external information. Recent works (Xu et al., 2024) have studied how models behave under such knowledge conflicts, but most analyses have focused on analyzing or controlling the behavior of already-pretrained models (Ortu et al., 2024; Yu et al., 2023; Li et al.), without examining how training conditions shape arbitration. However, it is essential to understand during pretraining what factors determine how a model uses and arbitrates between its two knowledge sources, so as to avoid discovering undesirable arbitration behaviors only after pretraining has consumed substantial resources.

Determining the appropriate knowledge source for a model is often challenging, given the variable provenance and reliability of in-context information. Our work therefore defines a robust arbitration strategy based solely on the internal signals of the model, without considering external factors. We define this strategy by two principles: (1) for high-confidence, well-memorized knowledge, the

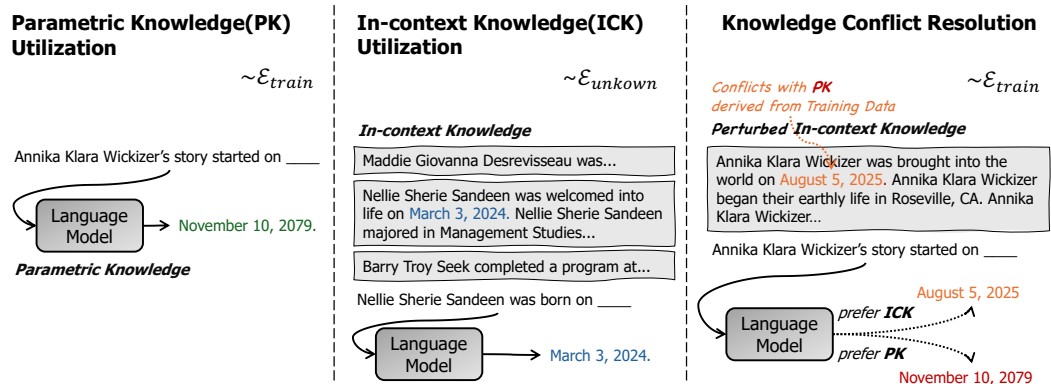

Figure 1: Three knowledge utilization scenarios. **Left:** parametric knowledge utilization where the model recalls knowledge encoded in its parameters and answers queries about entities seen during training. **Middle:** in-context knowledge utilization where the model extracts and uses knowledge provided only in the prompt and is evaluated on novel entities not seen during training. **Right:** knowledge conflict resolution where the model is queried about trained entities while the context provides conflicting information, and responses reveal the preference between parametric knowledge and in-context knowledge.

model should follow its parametric knowledge, even when faced with conflicting in-context knowledge; and (2) for novel or unfamiliar information, the model should follow the provided in-context knowledge. This behavior, which mirrors patterns in human cognition (Koriat, 2011) and has been observed in modern large language models (Wu et al., 2024), motivates our investigation of the training factors that produce it. Consequently, our study is guided by two central research questions: (**RQ1**) What training conditions enable a model to develop distinct capabilities for the use of parametric and in-context knowledge? (**RQ2**) What specific characteristics of the training corpus induce the model to adopt a robust arbitration strategy between these two sources?

To answer these questions, we conduct controlled experiments, training transformer-based language models from scratch on synthetic biographies (Allen-Zhu & Li, 2024a;b; Zucchet et al., 2025). This framework enables precise manipulation of training conditions while isolating knowledge utilization from other confounding factors. Prior work (Allen-Zhu & Li, 2024a) shows the insights from this setup can transfer to real-world models (Touvron et al., 2023), which allows us to explore a wide range of training configurations efficiently. Building on earlier studies (Zucchet et al., 2025) that focused only on parametric knowledge, we systematically investigate the interaction between parametric and in-context knowledge. During training with varied conditions, we evaluate the model's performance across three key knowledge utilization scenarios. First, we measure **parametric knowledge utilization** by the model's ability to recall learned entity attributes from its parameters. Second, we assess **in-context knowledge utilization** by its capacity to extract and use knowledge from the context for novel entities that are not in the training data. Finally, we examine **knowledge conflict resolution** by observing which source the model follows when known entities are paired with perturbed contexts, where the in-context knowledge conflicts with the model's parametric knowledge (Figure 1).

Our experiments led to the following findings: **Intra-document repetition** of facts is critical for the simultaneous emergence of both parametric knowledge and in-context knowledge utilization capabilities, and this in-context knowledge utilization ability emerges much earlier (Section 3.1). In addition, **a small degree of factual inconsistency** within a document encourages the model to favor its more confident parametric knowledge when conflicts arise, although during early training it initially prefers in-context knowledge (Section 3.2). Moreover, **a skewed frequency distribution of knowledge**, where long-tailed knowledge exists, preserves the model's ability to use in-context knowledge for unfamiliar entities. When these three conditions co-occur, they produce the desired arbitration pattern: the model defaults to parametric knowledge for well-learned entities but readily relies on in-context knowledge for rare or novel ones (Section 3.3). We validate these results on open-source, confirming that our findings extend beyond the synthetic setting (Section 4.1) and also post-training (Section 4.2).

These results have critical implications for pretraining large language models for retrieval augmented generation. We find that data characteristics often seen as defects, such as modest inconsistencies and a skewed knowledge distribution, are actually beneficial features for developing models that can intelligently arbitrate between learned knowledge and new, in-context information. A direct implication of this finding is that preprocessing steps like aggressive cleaning, deduplication, and data balancing may inadvertently impair a model's robust knowledge arbitration strategy.

## 2 DATASET AND SETUP

In this section, we describe our experimental framework. We first introduce our synthetic biographies dataset (Section 2.1), which enables precise control over knowledge distribution. We then detail our training setup (Section 2.2), followed by our evaluation framework (Section 2.3) that measures parametric knowledge utilization, in-context knowledge utilization, and conflict resolution behavior.

### 2.1 SYNTHETIC BIOGRAPHIES DATASET

We construct a synthetic biographies dataset following prior work (Allen-Zhu & Li, 2024a; Zucchet et al., 2025) (See details in Appendix A). Specifically, we generate synthetic biographical profiles, where each profile contains four attributes: `birth_date`, `birth_city`, `university`, and `major`. For each profile, we sample 7 distinct templates from a finite pool for each attribute. We use 6 templates to create training paragraphs with randomized attribute ordering, reserving 5 paragraphs for training and 1 for evaluation context. The remaining template for each attribute serves as test probes, which are cloze-style sentences designed to elicit the attribute values. Figure 9 illustrates the dataset structure. This deliberate separation ensures that the sentences used for training, context, and testing are never identical, compelling the model to utilize its parametric or in-context knowledge rather than relying on simple sequence memorization or repetition.

### 2.2 TRAINING SETUP

We train an 8-layer decoder-only Transformer language model from scratch (Vaswani et al., 2017), adopting the detailed hyperparameters(Table 5) from prior work (Zucchet et al., 2025). For the training entity set ($\mathcal{E}_{\text{train}}$), we use profiles of $50k$ entities, and for the unknown entity set ($\mathcal{E}_{\text{unknown}}$), we use other $50k$ entity profiles that are unseen during training. Using the training paragraphs of $e \in \mathcal{E}_{\text{train}}$ from Section 2.1, we assemble documents according to the variants described below and use the resulting collection as the training corpus.

### 2.3 EVALUATION SETUP

During training, we periodically evaluate the model at each checkpoint under three knowledge utilization scenarios: parametric knowledge utilization, in-context knowledge utilization, and knowledge conflict resolution, to measure the model's ability to utilize knowledge, as illustrated in Figure 1. We evaluate using the exact-match accuracy of the attributes generated by the model for the given input in each scenario. For each scenario, we randomly sample a set of $k$ entities for evaluation, and in our experiments, we set $k = 200$.

**Parametric Knowledge Utilization** This scenario measures the model's ability to utilize knowledge stored in its parameters. We evaluated this on entities seen during training, $e \sim \mathcal{E}_{\text{train}}$. The accuracy of parametric knowledge utilization is defined as $\text{Acc}_{\text{PKU}} = \mathbb{E}_{e \sim \mathcal{E}_{\text{train}}} \left[ \frac{1}{|A_e|} \sum_{a \in A_e} \mathbf{1}\{M(p_a) = v_a\} \right]$, where $A_e$ is the set of attributes of entity $e$, $p_a$ is the test probe for attribute $a$, $v_a$ is the ground-truth value, and $M(\cdot)$ is the model output.

**In-Context Knowledge Utilization** This scenario evaluates whether the model can utilize the knowledge provided only at inference time. We evaluated this on novel entities not seen during training, i.e., $e \sim \mathcal{E}_{\text{unknown}}$. For each unseen entity $e$, we construct a context $C$ by concatenating $C_e$ with paragraphs from two other random unseen entities, followed by shuffling. The accuracy of in-context knowledge utilization is defined as $\text{Acc}_{\text{ICKU}} = \mathbb{E}_{e \sim \mathcal{E}_{\text{unknown}}} \left[ \frac{1}{|A_e|} \sum_{a \in A_e} \mathbf{1}\{M(C, p_a) = v_a\} \right]$.

**Real World Documents**

WIKIPEDIA
The Free Encyclopedia

**Albert Einstein**[a] (14 March 1879 – 18 April 1955) was a German-born theoretical physicist who is best known for developing the theory of relativity. Einstein also made important contributions to quantum theory.[1][5] His mass–energy equivalence formula $E = mc^2$, which arises from special relativity, has been called "the world's most famous equation".[6] He received the 1921 Nobel Prize in Physics for *his services to theoretical physics, and especially for his discovery of the law of the photoelectric effect.*[7] Born in the German Empire, Einstein moved to Switzerland in 1895, forsaking his German citizenship (as a subject of the Kingdom of Württemberg)[note 1] the following year. In 1897, at the age of seventeen, he enrolled in the mathematics and physics …

☐ : Tokens can be predicted with **PK**

☐ : Tokens can be predicted with **PK** or **ICK**

**Our Synthetic Documents**

SINGLE

Annika Klara Wickizer was welcomed into the world in Roseville, CA.
Annika Klara Wickizer celebrates their birthday on November 10, 2079.
Annika Klara Wickizer earned qualifications in Information Systems.
Annika Klara Wickizer pursued higher education at Drew University.

REPEATED

Annika Klara Wickizer was welcomed into the world in Roseville, CA.
Annika Klara Wickizer celebrates their birthday on November 10, 2079.
Annika Klara Wickizer earned qualifications in Information Systems.
Annika Klara Wickizer pursued higher education at Drew University.
...
Annika Klara Wickizer first opened their eyes in Roseville, CA. Annika Klara Wickizer received their diploma from Drew University. Annika Klara Wickizer was welcomed into life on November 10, 2079. Annika Klara Wickizer was educated in the field of Information Systems
...

Figure 2: An example of intra-document repetition of key attributes (e.g., German, Physics) for a single entity, alongside our synthetic training-corpus variants. SINGLE uses one paragraph per entity and thus encourages reliance on parametric knowledge; REPEATED places two paraphrased paragraphs about the same entity in one document, allowing later mentions to leverage in-context knowledge or parametric knowledge.

**Knowledge Conflict Resolution** This scenario evaluates whether the model follows parametric knowledge (i.e., outputs the original training values) or in-context knowledge (i.e., outputs the values given in the perturbed context). For each training entity $e \sim \mathcal{E}_{\text{train}}$, we construct a perturbed context $C'_e$ by randomly altering two attributes (birth_date, university). Preference for parametric knowledge is defined as $\text{Pref}_{\text{PK}} = \mathbb{E}_{e \sim \mathcal{E}_{\text{train}}} \left[ \frac{1}{|A'_e|} \sum_{a \in A'_e} \mathbf{1}\{M(C'_e, p_a) = v_a\} \right]$, and preference for in-context knowledge as $\text{Pref}_{\text{ICK}} = \mathbb{E}_{e \sim \mathcal{E}_{\text{train}}} \left[ \frac{1}{|A'_e|} \sum_{a \in A'_e} \mathbf{1}\{M(C'_e, p_a) = v'_a\} \right]$, where $A'_e$ is the set of perturbed attributes, $v_a$ denotes the original parametric value from training, and $v'_a$ the conflicting value specified in $C'_e$.

## 3 EXPERIMENTS

In this section, we investigate the training conditions that enable models to develop distinct knowledge utilization capabilities (**RQ1**) and adopt robust arbitration strategies (**RQ2**). We begin by examining how intra-document repetition enables the emergence of both parametric and in-context knowledge utilization (Section 3.1). We then analyze how factual inconsistency noise influences the model's preference between conflicting knowledge sources (Section 3.2). Finally, we explore how skewed knowledge distribution preserves in-context knowledge utilization for unfamiliar entities while maintaining robust arbitration for well-learned ones (Section 3.3). Our findings reveal that these three factors must co-occur to produce the desired arbitration behavior.

### 3.1 EFFECTS OF INTRA-DOCUMENT REPETITION

Our first experiment addresses what training conditions enable a model to develop distinct capabilities for parametric and in-context knowledge utilization. We hypothesize that *intra-document repetition*—a common feature of real-world text where key attributes are repeated within documents (Figure 2)—is critical for simultaneously developing both capabilities. During next-token prediction, the first mention of an attribute lacks in-document context, requiring parametric knowledge, while later mentions allow the model to leverage earlier context.

**Training corpus variants** We construct two corpus variants to test this hypothesis:

- SINGLE: Each training document contains exactly one training paragraph about a single entity. Attributes appear once per document, so the model cannot rely on in-context knowledge; predictions must be supported by parametric knowledge.

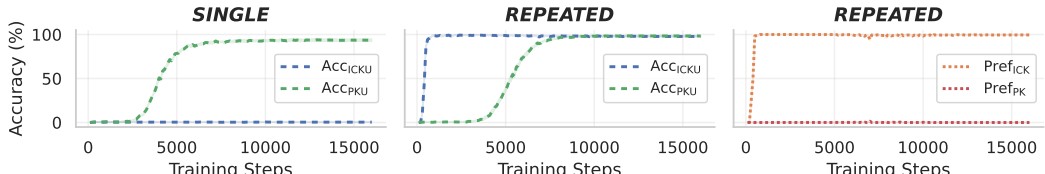

Figure 3: Accuracy of parametric knowledge utilization ($Acc_{PKU}$) and in-context knowledge utilization ($Acc_{ICKU}$) across training steps and knowledge conflict resolution results. (**Left**) The SINGLE corpus shows delayed parametric knowledge utilization with no in-context knowledge activation. (**Middle**) The REPEATED corpus induces early in-context knowledge utilization followed by parametric knowledge utilization. (**Right**) The REPEATED model consistently prefers in-context knowledge under conflict.

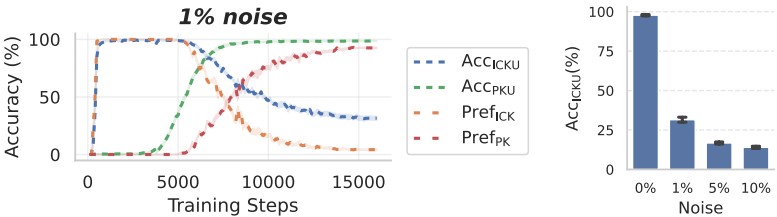

Figure 4: (**Left**) Training dynamics of $Acc_{ICKU}$, $Acc_{PKU}$, $Pref_{ICK}$, and $Pref_{PK}$ when trained on the REPEATED corpus with 1% noise. Even a small amount of noise induces a phase shift toward parametric knowledge preference as parametric knowledge utilization stabilizes. (**Right**) $Acc_{ICKU}$ at the end of training across different noise levels.

- REPEATED: We construct documents in which each attribute of an entity appears twice, such that the first mention necessarily relies on parametric knowledge while the second mention provides an opportunity for the model to use either parametric knowledge or in-context knowledge. To avoid trivial copying based solely on previously mentioned attribute types regardless of the subject, we mix multiple entities within each document: specifically, we sample two paraphrased paragraphs for each of three distinct entities and shuffle all six paragraphs to form a single training document.

**Results**   Figure 3 (left and middle) shows that the SINGLE corpus develops only parametric knowledge utilization midway through training, with no in-context knowledge activation. In contrast, the REPEATED corpus enables both capabilities, with in-context knowledge emerging earlier than parametric knowledge. This ordering aligns with prior work (Zucchet et al., 2025): parametric knowledge requires complex attention circuits connecting subject tokens to attributes in key-value format (Meng et al., 2022; Geva et al., 2023), learned gradually through backpropagation. In-context knowledge relies on simpler induction heads (Olsson et al., 2022) that copy attribute tokens, reflecting general patterns rather than entity-specific knowledge, thus emerging earlier.

These results confirm that parametric knowledge utilization does not automatically enable in-context knowledge utilization. Rather, intra-document repetition creates a training environment where in-context knowledge emerges first, followed by gradual parametric knowledge activation.

### 3.2 EFFECTS OF FACTUAL INCONSISTENCY NOISE WITHIN A DOCUMENT

**Models trained without noise over-rely on in-context knowledge**   The model trained on the REPEATED corpus successfully leverages both parametric and in-context knowledge. However, when evaluating knowledge conflict resolution, we find that the model invariably follows in-context knowledge, even after parametric knowledge utilization reaches nearly 100% accuracy (Figure 3 right). This preference persists despite the model being highly confident in its parametric knowledge for training entities, as measured by entropy and target token probability (Table 1). This tendency to over-rely on external context deviates from the robust arbitration strategy observed in real-world

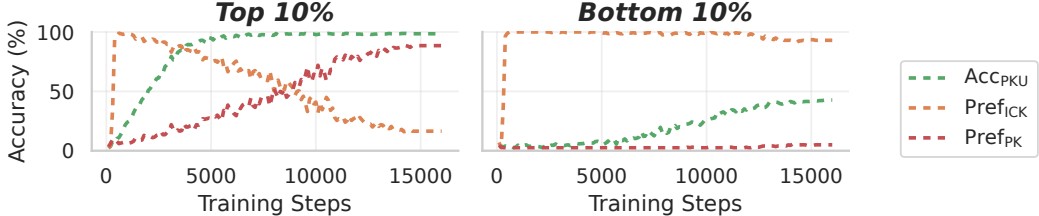

Figure 5: $\text{Acc}_{\text{PKU}}$, $\text{Pref}_{\text{ICK}}$, and $\text{Pref}_{\text{PK}}$ for the top 10% (high-frequency) and bottom 10% (low-frequency) entities in the training corpus. For high-frequency entities, $\text{Pref}_{\text{ICK}}$ is initially higher but gradually yields to $\text{Pref}_{\text{PK}}$; for low-frequency entities, $\text{Pref}_{\text{ICK}}$ remains consistently higher.

large language models (Yu et al., 2023; Wu et al., 2024), which tend to prefer parametric knowledge for frequently encountered or high-confidence information.

**Inconsistency noise induces robust parametric knowledge preference**  We hypothesize that the inevitable presence of noise in web corpora—such as typos, factual errors, or conflicting opinions—introduces small inconsistencies that prevent models from blindly following in-context knowledge. To test this, we train models on the REPEATED corpus with factual inconsistency noise. Specifically, for each entity mentioned in two paragraphs within a document, we perturb the `birth_date` and `major` values only in the leading paragraph with probability $p \in \{1\%, 5\%, 10\%\}$, leaving the later paragraph unchanged (Figure 10).

As shown in Figure 4 (left), early in training, the model prefers in-context knowledge in conflicts. However, as parametric knowledge utilization stabilizes, the model

Table 1: Target token probability and entropy measured at the last token of the test probe for entities in $\mathcal{E}_{\text{train}}$ and $\mathcal{E}_{\text{unknown}}$.

|  | $\mathcal{E}_{\text{train}}$ | $\mathcal{E}_{\text{unknown}}$ |
|---|---|---|
| **w/o noise** | | |
| Target prob. | 0.998 | 0.024 |
| Entropy (nats) | 0.011 | 0.955 |
| **w/ 1% noise** | | |
| Target prob. | 0.997 | 0.034 |
| Entropy (nats) | 0.016 | 1.236 |

gradually shifts toward preferring parametric knowledge. Remarkably, even 1% noise is sufficient to induce this phase shift. This demonstrates that a small degree of inconsistency enables the model to robustly favor high-confidence parametric knowledge over conflicting in-context information.

**Trade-off: degradation of in-context knowledge utilization**  However, this comes at a cost: in-context knowledge utilization performance degrades, with the decline increasing at higher noise levels (Figure 4 right). Importantly, this degradation is not due to the model hallucinating knowledge for unfamiliar entities. As Table 1 shows, the model maintains high confidence for training entities while showing low confidence for unknown entities, indicating it can still distinguish what it knows from what it does not.

Despite recognizing its lack of knowledge about $\mathcal{E}_{\text{unknown}}$ entities (as evidenced by low target probability and high entropy in Table 1), the model fails to utilize available in-context knowledge. Detailed attention pattern analysis (Appendix F) reveals the underlying mechanism: even for unknown entities about which the model is highly uncertain, attention circuits preferentially retrieve information from subject name tokens rather than from target attribute tokens in the context. This suggests that the model has shifted toward using parametric knowledge retrieval mechanisms even in situations where parametric knowledge is unavailable, effectively forgetting how to leverage in-context knowledge despite recognizing its own uncertainty.

In summary, while a small amount of inconsistency noise enables robust parametric knowledge preference for high-confidence knowledge, it also leads to over-reliance on parametric knowledge mechanisms, ultimately degrading in-context knowledge utilization capabilities.

### 3.3 EFFECTS OF SKEWED KNOWLEDGE DISTRIBUTION

**Skewed Knowledge Distribution Preserves in-context knowledge Utilization on Unfamiliar Knowledge**  We hypothesize that to prevent the degradation of in-context knowledge utilization for unfamiliar knowledge, as observed in Section 3.2, the training data must continually expose the

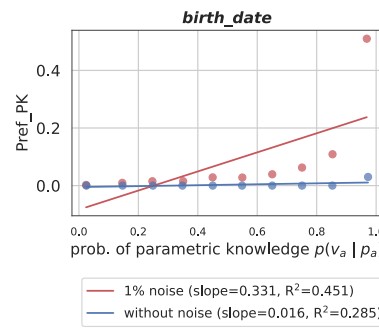 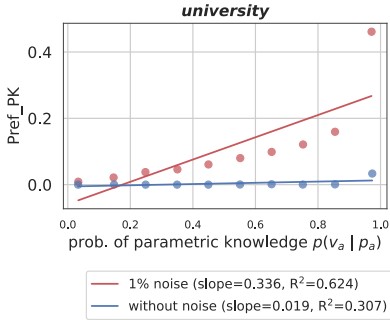

Figure 6: Correlation between parametric knowledge confidence and parametric knowledge preference under conflict for two attributes (`birth_date` and `university`). We bin entities into 10 groups based on their parametric knowledge probability $p(v_a|p_a)$ and measure $\text{Pref}_{\text{PK}}$ for each bin. (**Red**) Model trained with 1% noise shows strong positive correlation (slope $\approx 0.33$, $R^2 \approx 0.45$-0.62), indicating that higher confidence leads to stronger parametric knowledge preference. (**Blue**) Model trained without noise shows near-zero correlation (slope $\approx 0.02$, $R^2 \approx 0.29$-0.31), failing to follow parametric knowledge even when highly confident.

model to information that cannot be recalled purely from parametric knowledge. In other words, knowledge from long-tailed knowledge should appear repeatedly so that in-context knowledge utilization remains active and does not degenerate. To test this, we constructed a REPEATED corpus where entities are sampled according to a Zipfian distribution(Zipf, 2012)[1](with small inconsistency noise as in Section 3.2). As shown in Table 2, training on this corpus yielded substantially less degradation in in-context knowledge utilization compared to training on a corpus with a uniform knowledge distribution.

We further evaluated preference measures across the top 10% and bottom 10% of entities by frequency (Figure 5) to examine under knowledge conflicts which knowledge the model follows for entities it saw frequently during training versus those it saw infrequently.

For high-frequency entities, the model initially preferred in-context knowledge but gradually shifted toward a robust reliance on parametric knowledge, as when trained under a uniform distribution. In contrast, for low-frequency entities, the model continued to prefer in-context knowledge. Importantly, this is not simply due to an inability to recall these entities from parametric knowledge. Because $\text{Acc}_{\text{PKU}}$ exceeds $\text{Pref}_{\text{PK}}$, the model retains parametric knowledge for some of these entities, yet in conflict settings it still prefers in-context knowledge over parametric knowledge. This tendency to rely on in-context knowledge for low-frequency entities supports our hypothesis that the model continues to use in-context knowledge when unfamiliar knowledge arises during training, thereby preventing degradation of in-context knowledge utilization capability.

Table 2: $\text{Acc}_{\text{ICKU}}$ at the end of training on uniform vs. Zipfian ($\alpha = 1$) corpora. The Zipfian column shows the change relative to the corresponding uniform value in parentheses.

| Noise | $\text{Acc}_{\text{ICKU}}$ (%) | |
|---|---|---|
| | Uniform | Zipfian |
| 1% | 31.5 | 84.0 (+52.5) |
| 5% | 16.8 | 63.9 (+47.1) |
| 10% | 14.1 | 57.4 (+43.3) |

**Skewness Alone Fails to Build Robust Parametric Knowledge Preference** We investigate whether internal confidence in parametric knowledge aligns with conflict resolution behavior in models trained with different corpus characteristics. Specifically, we examine whether models prefer parametric knowledge when they are confident about it, as measured by the probability of the correct answer during parametric knowledge utilization, $p(v_a|p_a)$. For each model, we divide entities into 10 bins based on their parametric knowledge probability and compute $\text{Pref}_{\text{PK}}$ for each bin, then analyze the correlation between confidence and preference.

Figure 6 shows the results for two attributes (`birth_date` and `university`). For the model trained with 1% inconsistency noise on a Zipfian corpus (red), we observe a strong positive corre-

[1]Zipfian distribution: $P(r) = r^{-\alpha}/\sum_{k=1}^{N} k^{-\alpha}$, where $r$ is the rank (1 = most frequent).

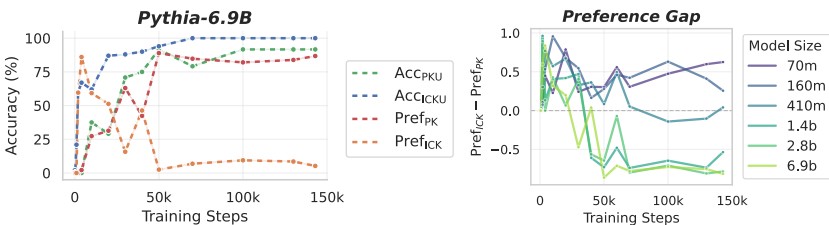

Figure 7: Training dynamics of knowledge utilization and conflict resolution in Pythia models. (**Left**) $\text{Acc}_{\text{ICKU}}$, $\text{Acc}_{\text{PKU}}$, $\text{Pref}_{\text{ICK}}$, and $\text{Pref}_{\text{PK}}$ across training steps for Pythia-6.9B. (**Right**) Preference gap ($\text{Pref}_{\text{ICK}} - \text{Pref}_{\text{PK}}$) across different model sizes, showing a consistent pattern of initial increase followed by decline as training progresses.

lation: as parametric knowledge confidence increases, so does the preference for parametric knowledge under conflict . This demonstrates that the model has developed a confidence-calibrated arbitration strategy, where it relies on parametric knowledge when confident and defers to in-context knowledge when uncertain.

In contrast, the model trained without noise (blue) shows virtually no correlation. Even when the model exhibits high confidence in its parametric knowledge (high $p(v_a|p_a)$), it fails to prefer parametric knowledge under conflict, instead consistently following in-context knowledge regardless of its internal confidence. This indicates that while a skewed knowledge distribution preserves in-context knowledge utilization for unfamiliar entities, it is insufficient on its own to develop a robust, confidence-aligned arbitration strategy. Only the combination of skewed distribution and modest inconsistency noise produces models that intelligently arbitrate based on their internal confidence.

## 4 DISCUSSION

### 4.1 VALIDATION ON REAL-WORLD MODELS

We have shown through controlled experiments that (i) intra-document repetition enables the joint emergence of parametric and in-context knowledge utilization, (ii) a small degree of factual inconsistency noise within a document biases conflict resolution toward confident parametric knowledge, and (iii) distributional skew with long-tailed knowledge preserves in-context utilization for unfamiliar entities. Because these properties arise naturally in web corpora, we test whether the same dynamics appear in real-world open-source LLMs.

Using the publicly released checkpoints of the PYTHIA model suite (Biderman et al., 2023), we evaluate parametric utilization, in-context utilization, and preference under knowledge conflict at each checkpoint (evaluation details in Figure 14 and Appendix E). As shown in Figure 7 (left), PYTHIA-6.9B exhibits dynamics consistent with our synthetic experiments: in-context utilization emerges earlier than parametric utilization, the model initially prefers in-context knowledge under conflict but gradually shifts toward parametric knowledge, while maintaining high $\text{Acc}_{\text{ICKU}}$ for novel entities throughout training.

To examine how this phase transition varies across model scales, we analyze the preference gap ($\text{Pref}_{\text{ICK}} - \text{Pref}_{\text{PK}}$) for models ranging from 70M to 6.9B parameters (Figure 7 right). All models exhibit the consistent pattern of initial increase followed by decline. Notably, larger models show stronger parametric knowledge preference at the end of training, with the preference gap approaching $-1$ for the largest models, consistent with prior observations that larger models tend to rely more heavily on their parametric knowledge (Yu et al., 2023). This trend can be attributed to larger models developing parametric knowledge more rapidly and robustly, leading to higher confidence in their internal knowledge and consequently stronger preference for it when conflicts arise.

These results indicate that repetition, small amounts of inconsistency noise, and skewed knowledge distributions in web-scale data naturally reproduce the dynamics observed in our synthetic setting, suggesting that our findings extend broadly to practical language model pretraining scenarios.

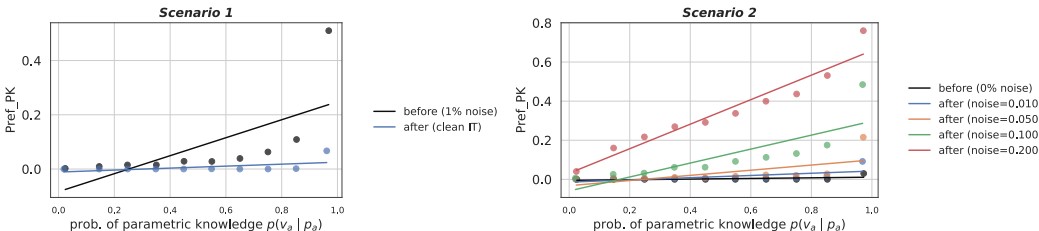

Figure 8: Correlation between parametric knowledge confidence and preference under conflict before and after post-training. (**Left**) Scenario 1: A model pretrained with 1% noise shifts toward near-zero correlation after clean post-training, mirroring the behavior of models pretrained without noise. (**Right**) Scenario 2: A model pretrained without noise develops increasing positive correlation as post-training noise increases, with higher noise levels producing stronger confidence-aligned parametric knowledge preference.

## 4.2 OVERWRITING KNOWLEDGE ARBITRATION STRATEGY THROUGH POST-TRAINING

Our findings reveal how training corpus characteristics shape knowledge arbitration strategies during pretraining. A natural question arises: can these strategies be modified after pretraining through post-training procedures.

We first examine whether instruction tuning affects arbitration behavior in real-world models. We evaluate two pairs of base and instruction-tuned models: Pythia-6.9B, and OLMo-7B (Groeneveld et al., 2024). As shown in Table 3, both base models exhibit strong parametric knowledge preference, consistent with our observations in Section 4.1. However, after instruction tuning on Tulu dataset(Wang et al., 2023), both models show a reversal: parametric knowledge preference drops while in-context knowledge preference increases to over 70%. This suggests that instruction tuning, which typically involves relatively clean and well-structured data, can significantly alter the arbitration strategy established during pretraining.

Table 3: Knowledge conflict resolution preferences before and after instruction tuning(IT) for Pythia-6.9B and OLMo-7B. Both models show a shift from parametric knowledge preference to in-context knowledge preference after IT.

|      | Pythia-6.9B | | OLMo-7B | |
| --- | --- | --- | --- | --- |
|      | $\text{Pref}_{\text{PK}}$ | $\text{Pref}_{\text{ICK}}$ | $\text{Pref}_{\text{PK}}$ | $\text{Pref}_{\text{ICK}}$ |
| Base | 0.8677 | 0.0525 | 0.5507 | 0.3894 |
| IT   | 0.1829 | 0.7771 | 0.2137 | 0.7155 |

**Controlled post-training validates the role of noise in arbitration** To test whether our observations extend to post-training, we conduct controlled experiments on synthetic models using answer-only loss with 1,000 entities trained for 500 steps. We examine two scenarios with Zipfian-pretrained models (Figure 8): (1) a model pretrained with 1% noise is post-trained on clean data, and (2) a model pretrained without noise is post-trained with varying noise levels ($p \in \{1\%, 5\%, 10\%, 20\%\}$).

We analyze the confidence-preference correlation by binning entities based on parametric knowledge probability and measuring $\text{Pref}_{\text{PK}}$ for each bin. In Scenario 1 (left), the noised-pretrained model's positive correlation flattens to near-zero after clean post-training, indicating a shift toward unconditional in-context knowledge preference regardless of confidence. In Scenario 2 (right), the clean-pretrained model develops increasingly strong positive correlations as noise levels increase, with 20% noise producing the steepest slope. These bidirectional shifts demonstrate that arbitration strategies can be systematically modified through post-training data characteristics, validating that our findings extend beyond pretraining.

**Implications** These results demonstrate that our findings on corpus characteristics extend beyond pretraining to post-training scenarios. Simply adjusting the factual consistency of post-training data is sufficient to systematically reshape arbitration strategies, whether toward trusting external context or relying on internal knowledge.

## 5 RELATED WORKS

Large language models rely on both parametric and in-context knowledge (Lewis et al., 2021; Mallen et al., 2022; Ram et al., 2023; Shi et al., 2023). Recent studies show that model preferences in conflicts depend on confidence and training frequency (Wu et al., 2024; Yu et al., 2023), and can be steered through attention manipulation or contrastive decoding (Li et al.; Yu et al., 2023; Sun et al., 2025; Jin et al., 2024). However, these works mainly focus on post-pretraining behavior and provide limited insight into how the ability to handle the two sources develops during training. A complementary line of research investigates training dynamics of language models using synthetic datasets (Allen-Zhu & Li, 2024a;b; Zucchet et al., 2025), enabling controlled studies of how models acquire and store knowledge. While these studies illuminate the formation of parametric knowledge, they do not address the simultaneous development of in-context utilization (Olsson et al., 2022) or the dynamics of conflict resolution. We bridge these directions by conducting the first systematic analysis of how parametric and in-context knowledge utilization co-emerge and interact during pre-training.

In parallel, some studies (Chan et al., 2022) that investigate in-context learning with transformer classifiers on Omniglot datasets (Lake et al., 2019) report that a skewed data distribution is required for in-context and in-weight (parametric) learning to co-exist. In contrast, our results show co-existence even under a uniform distribution. We attribute this difference to the task setup: those classification tasks can be solved solely from exemplars provided in context, predicting only a class token conditioned on a query, whereas a language model performs next-token prediction for every data sequence and thus must rely on parametric knowledge for most initial content, which more strongly incentivizes reliance on parametric knowledge than in exemplars-conditioned classification. Building on this distinction, we study knowledge-conflict scenarios (Neeman et al., 2022) and settings closer to how real-world language models are trained (Brown et al., 2020).

## 6 CONCLUSION

We present the first systematic analysis of how training corpus characteristics shape parametric and in-context knowledge utilization in language models. Through controlled experiments on synthetic biographies and validation on the Pythia model suite, we identify three critical factors that must co-occur for robust arbitration: (1) intra-document repetition enables both knowledge utilization capabilities to emerge, (2) small degrees of factual inconsistency induce preference for high-confidence parametric knowledge, and (3) skewed knowledge distributions preserve in-context knowledge utilization for unfamiliar entities. Furthermore, we demonstrate that these arbitration strategies can be systematically modified through post-training by adjusting data consistency.

## REPRODUCIBILITY STATEMENT

We describe the dataset construction process in detail in Section 2 and Appendix A. The hyperparameters and model configuration used in our experiments are provided in Appendix B. Furthermore, we will release code for experiments publicly. All experiments are implemented using the `HuggingFace TRL` library[2] and conducted on a single NVIDIA A100 GPU. Each training run requires approximately 4–6 hours.

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

## A  SYNTHETIC BIOGRAPHIES DATASET CONSTRUCTION

Following prior work (Allen-Zhu & Li, 2024a; Zucchet et al., 2025), we first construct $N$ synthetic person profiles. Each profile contains four attributes: `birth_date`, `birth_city`, `university`, and `major`. Names (first/middle/last) are sampled by randomly composing entries from a public name database.[3] For `birth_date`, we sample a date uniformly from 1900–2099. For `birth_city` and `university`, we sample from curated lists of 200 values each, and for `major` from a list of 100 values, all derived from Wikipedia.[4] For each attribute, we sample 7 distinct surface templates from a finite template pool. An example of templates for `birth_date` is shown below.

---

**An example of templates for `birth_date`**

1. `person was born on birth_date.`
2. `person came into the world on birth_date.`
3. `person entered this world on birth_date.`
4. `person was brought into the world on birth_date.`
5. `person took their first breath on birth_date.`
6. `person began their life journey on birth_date.`
7. `person celebrates their birthday on birth_date.`
8. `person first opened their eyes on birth_date.`
9. `person was welcomed into life on birth_date.`
10. `person arrived on birth_date.`
11. `person's story started on birth_date.`
12. `person was born to the world on birth_date.`
13. `person was delivered into the world on birth_date.`
14. `person was given life on birth_date.`
15. `person was welcomed into the world on birth_date.`
16. `person began their journey on Earth on birth_date.`
17. `person made their debut in the world on birth_date.`
18. `person became a part of the world on birth_date.`
19. `person was born into this life on birth_date.`
20. `person came to life on birth_date.`

---

We then create paragraphs containing each person's biography with a randomized attribute order as follows: using 6 of the templates, we generate six paragraphs per entity; five are reserved for training and one is used as the evaluation in-context paragraph. The remaining (seventh) template is held out as a closed-style test probe designed to elicit the target attribute. An illustration of the resulting dataset is shown in Figure 9.

## B  DETAILS ON TRAINING LANGUAGE MODELS

For our controlled experiments, we use a GPT-2–style decoder-only Transformer[5]. The model configuration is summarized in Table 4. Following Hoffmann et al. (2022), we adopt the settings used in Zucchet et al. (2025). The training hyperparameters are listed in Table 5.

---

[3]https://github.com/smashew/NameDatabases/tree/master/NamesDatabases
[4]https://en.wikipedia.org/wiki/
[5]https://huggingface.co/openai-community/gpt2

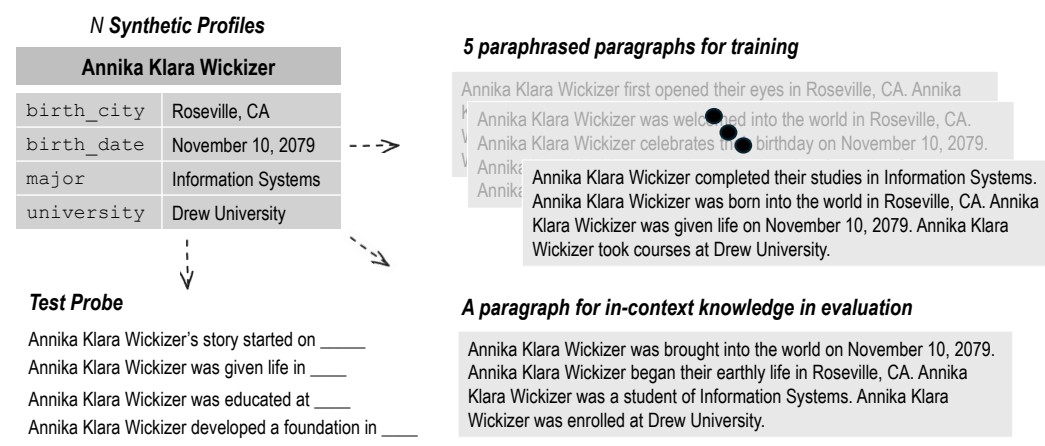

Figure 9: An example of the synthetic dataset. Each profile consists of four attributes (`birth_date`, `birth_city`, `university`, `major`), with paragraphs for training, a paragraph for in-context knowledge in evaluation, and test probes for eliciting the model to generate the attributes of each entity.

Table 4: Model architecture.

| Component | Value |
|---|---|
| Embedding dimension | 512 |
| Layers | 8 |
| Attention heads | 8 |
| FFN inner dimension | 2048 |
| Context length | 512 |

## C  EXAMPLE OF FACTUAL INCONSISTENCY NOISE WITHIN A DOCUMENT

Figure 10 illustrates a document from the REPEATED+MIX corpus in which factual inconsistency noise has been injected. The value highlighted in pink was injected as noise with some probability and therefore does not match the latter original value, "November 10, 2079."

## D  ADDITIONAL EXPERIMENTAL RESULTS

We further examine the training dynamics by systematically varying several factors. Unless otherwise noted, all experiments are conducted on the REPEATED+MIX corpus.

### D.1  EFFECT OF THE NUMBER OF TRAINING ENTITIES

Figure 11 compares REPEATED+MIX runs with 50k, 100k, and 200k training entities. With 50k entities, both in-context knowledge utilization ($Acc_{ICKU}$) and parametric knowledge utilization ($Acc_{PKU}$) emerge, with $Acc_{ICKU}$ activating earlier and $Acc_{PKU}$ following as training stabilizes. In contrast, for 100k and 200k entities, $Acc_{PKU}$ fails to rise: the model learns to use in-context knowledge but does not develop robust parametric utilization.

### D.2  EFFECT OF INTRA-DOCUMENT INCONSISTENCY NOISE

Figure 12 examines training dynamics under intra-document factual inconsistency levels of 1%, 5%, and 10%. Even 1% noise is sufficient to induce a phase shift in conflict-time preference: as $Acc_{PKU}$ stabilizes, the model transitions from preferring in-context knowledge ($Pref_{ICK}$) to preferring parametric knowledge ($Pref_{PK}$). Increasing noise accelerates this shift but also degrades

Table 5: Training hyperparameters.

| Hyperparameter | Value |
|---|---|
| Max training steps | 16,000 |
| Batch size | 128 |
| Learning rate | $4 \times 10^{-4}$ |
| Weight decay | 0.10 |
| LR scheduler | Cosine |
| Sequence length | 512 |
| Numerical precision | bfloat16 |

Annika Klara Wickizer was welcomed into the world in Roseville, CA. Annika Klara Wickizer celebrates their birthday on    **August 5, 1999**   . Annika Klara Wickizer earned qualifications in Information Systems. Annika Klara Wickizer pursued higher education at Drew University.

Dara Angila Honey was given life on April 6, 1978. Dara Angila Honey focused their academic efforts on Industrial. Dara Angila Honey entered this world in Indianapolis, IN. Dara Angila Honey achieved academic success at Fisk University.

Dara Angila Honey chose Industrial as their field of study. Dara Angila Honey completed a program at Fisk University. Dara Angila Honey was welcomed into life on April 6, 1978. Dara Angila Honey became a part of the world in Indianapolis, IN.

Annika Klara Wickizer first opened their eyes in Roseville, CA. Annika Klara Wickizer received their diploma from Drew University. Annika Klara Wickizer was welcomed into life on **November 10, 2079.** Annika Klara Wickizer was educated in the field of Information Systems.

Roselee Justine Woolem gained academic grounding in Business Analytics. Roselee Justine Woolem first opened their eyes in Phoenix, AZ. Roselee Justine Woolem studied at Hamilton College. Roselee Justine Woolem was brought into the world on August 12, 2083.

Roselee Justine Woolem entered this world on August 12, 2083. Roselee Justine Woolem majored in Business Analytics. Roselee Justine Woolem began their life in Phoenix, AZ. Roselee Justine Woolem developed expertise at Hamilton College.

Figure 10: Example of the document injected inconsistency noise

$Acc_{ICKU}$ at convergence, indicating over-reliance on parametric knowledge and a reduced ability to use in-context knowledge.

### D.3    EFFECT OF DISTRIBUTIONAL SKEW

Figure 13 examines training dynamics under Zipfian sampling with $\alpha \in \{0.5, 1.0, 2.0\}$. A near-uniform regime ($\alpha$=0.5) yields progressive degeneration of $Acc_{ICKU}$ over training, consistent with the model drifting toward parametric recall even for unfamiliar entities. An overly skewed regime ($\alpha$=2.0) produces undesirable dynamics—parametric utilization fails to activate—suggesting that extreme concentration of exposure undermines balanced capability growth. A moderate skew ($\alpha$=1.0) best preserves $Acc_{ICKU}$ for rare or novel entities while still supporting stable $Acc_{PKU}$ and a robust preference for parametric knowledge on frequently seen facts.

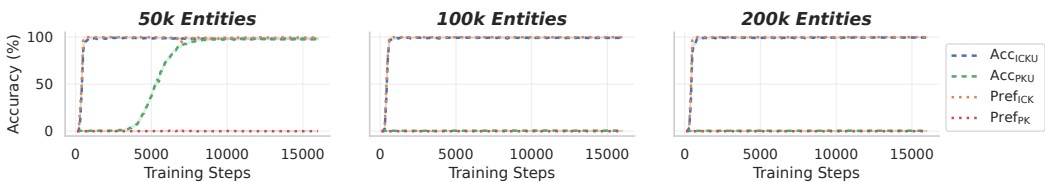

Figure 11: Training dynamics of $\mathrm{Acc_{ICKU}}$ and $\mathrm{Acc_{PKU}}$ under different numbers of training entities.

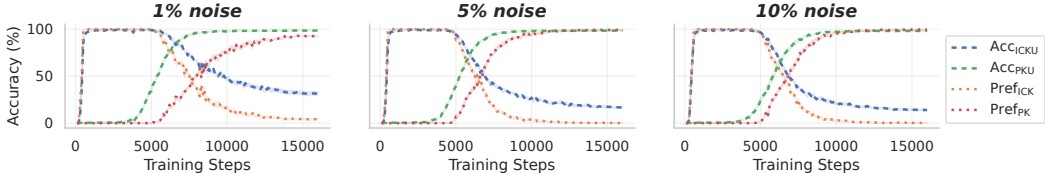

Figure 12: Training dynamics of $\mathrm{Acc_{ICKU}}$ and $\mathrm{Acc_{PKU}}$ under different levels of intra-document inconsistency noise.

## E    EXPERIMENTAL DETAILS FOR REAL-WORLD LARGE LANGUAGE MODELS

We adapt the scenarios used in our controlled experiments so that they can be applied to models trained on real web corpora. Since web corpora contain abundant information about countries and their capitals, we designate the set of training entities $\mathcal{E}_{\mathrm{train}}$ as *Real-World Countries* and evaluate whether the model can correctly predict their corresponding capital cities. To this end, we construct a *Real-World Country–Capital Set* based on the country–capital data pairs used in Hernandez et al. (2023). Using this dataset, we build question–answer style test probes as illustrated in Figure 14, and define the **Parametric Knowledge Utilization** scenario. We then measure $\mathrm{Acc_{PKU}}$ by checking whether the model's generations within 64 tokens contain the correct answer.

For the **In-Context Knowledge Utilization** scenario, we need to evaluate knowledge unseen during training. Therefore, we create 100 artificial country–capital pairs that do not exist in the real world, forming a *Synthetic Country–Capital Set*. As described in Section 2.3, we embed these pairs into a context and provide them to the model along with a test probe, measuring $\mathrm{Acc_{ICKU}}$ by verifying whether the correct answer appears within the first 64 generated tokens.

Finally, for **Knowledge Conflict Resolution**, we perturb the in-context knowledge by replacing the true answers in the *Real-World Country–Capital Set* with incorrect ones. We then provide these perturbed contexts together with the test probes and evaluate whether the model follows the perturbed in-context knowledge or the true answer. This allows us to measure $\mathrm{Pref_{ICK}}$ and $\mathrm{Pref_{PK}}$.

## F    ATTENTION PATTERN ANALYSIS

In this section, we provide additional analysis of attention patterns to understand the mechanisms underlying the degradation of in-context knowledge utilization observed in Section 3.2. Specifically, we investigate how the model trained on the REPEATED corpus with 1% inconsistency noise allocates attention during the training process, which allows us to indirectly examine the circuits used for parametric versus in-context knowledge utilization.

**Attention Patterns During In-Context Knowledge Utilization**    To understand why in-context knowledge utilization degrades when trained with inconsistency noise, we analyzed the attention patterns at the last token position of the test probe during in-context knowledge utilization for $\mathcal{E}_{\mathrm{unknown}}$ entities. Figure 15 shows the layer-wise sum of attention mass over the course of training. We distinguish between two types of attention targets: (1) name tokens in the test probe (shown in green), which are associated with parametric knowledge retrieval, and (2) target attribute tokens in the context (shown in blue), which are necessary for in-context knowledge utilization.

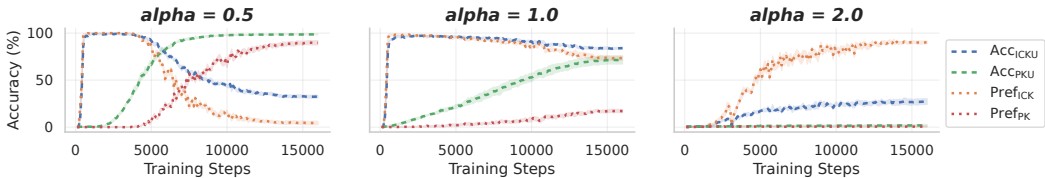

Figure 13: Training dynamics of $\text{Acc}_{\text{ICKU}}$ and $\text{Acc}_{\text{PKU}}$ as a function of the Zipf exponent $\alpha$.

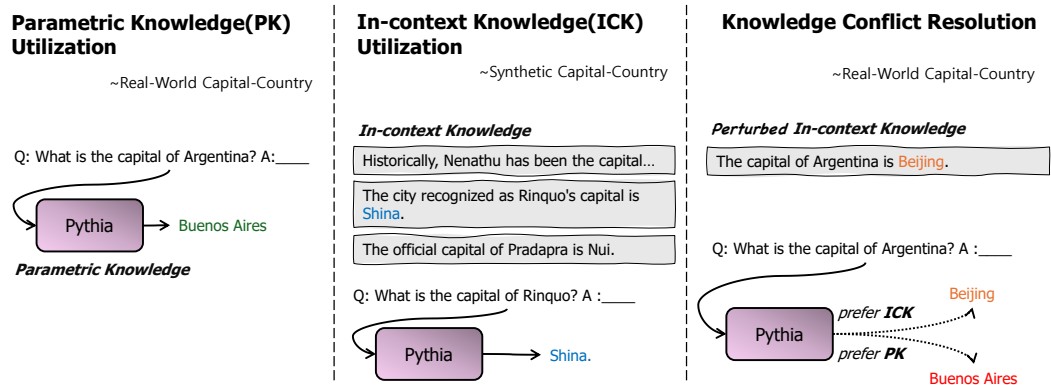

Figure 14: Three knowledge utilization scenarios in real-world large language models. **Left:** Parametric knowledge utilization, where the model recalls country–capital facts from real-world data that were encoded in its parameters during training. **Middle:** In-context knowledge utilization, where the model relies on synthetic country–capital pairs provided only in the context. **Right:** Knowledge conflict resolution, where the model is queried about real-world countries while the prompt supplies perturbed (incorrect) capitals, allowing us to examine whether the model prefers parametric knowledge or the perturbed in-context knowledge.

Our analysis reveals a clear temporal pattern: early in training, attention focuses heavily on target tokens in the context, which is consistent with successful in-context knowledge utilization. However, as training progresses and parametric knowledge utilization stabilizes, attention gradually shifts toward name tokens in the test probe. This shift is particularly notable because it occurs even when evaluating on $\mathcal{E}_{\text{unknown}}$ entities—entities for which the model has no parametric knowledge.

**Interpretation** Prior work (Meng et al., 2022) has established that recalling parametric knowledge requires retrieving information from relevant subject tokens through specific attention circuits (Zucchet et al., 2025; Geva et al., 2023). The observed shift in attention patterns suggests that, even for unknown entities, the model attempts to recall information from name tokens to utilize parametric knowledge, following the same mechanisms established for parametric knowledge utilization during training.

This finding provides crucial insight into the degradation of in-context knowledge utilization observed in Section 3.2. While the model can distinguish between entities it knows (high confidence, low entropy) and entities it does not know (low confidence, high entropy), as shown in Table 1, it appears to have forgotten how to leverage in-context knowledge. The attention patterns reveal that the model has not lost the ability to recognize unfamiliar entities, but rather has shifted to preferentially using parametric knowledge retrieval circuits even in situations where in-context knowledge would be more appropriate.

In other words, the presence of inconsistency noise during training leads the model to develop a strong bias toward parametric knowledge utilization mechanisms. This bias persists even when parametric knowledge is unavailable, causing the model to attempt parametric retrieval for unknown entities rather than falling back on available in-context information. This mechanistic understanding helps explain why skewed knowledge distribution (Section 3.3) is necessary to preserve in-context

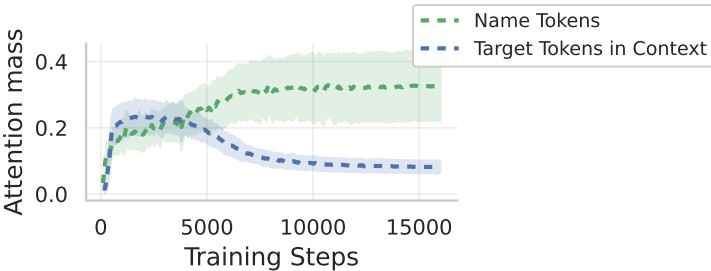

Figure 15: Changes in the layer-wise sum of attention mass at the last token of the test probe when the model trained with 1% noise performs in-context knowledge utilization for $\mathcal{E}_{\text{unknown}}$ entities. Green indicates the attention allocated to name tokens in the test probe, while blue indicates the attention allocated to target tokens in the context. As training progresses, attention gradually shifts from context target tokens to name tokens, indicating a transition from in-context knowledge utilization circuits to parametric knowledge retrieval circuits.

knowledge utilization: the continuous presence of unfamiliar entities in the training distribution prevents the complete abandonment of in-context knowledge circuits.

## G  THE USE OF LARGE LANGUAGE MODELS

We used large language models solely to aid and polish the writing of this paper, including tasks such as grammar correction, wording refinement, and minor stylistic edits.

