# OpenReview forum: "Training Dynamics of Parametric and In-Context Knowledge Utilization in Language Models"
_ICLR.cc/2026/Conference — ICLR 2026 Conference Withdrawn Submission_

### Official Review · Reviewer_77zM · 2025-10-14

**Soundness:** 2
**Presentation:** 3
**Contribution:** 2
**Rating:** 4
**Confidence:** 4

**Summary:**

This paper presents a controlled study on how pretraining data characteristics influence the development of a language model's ability to utilize and arbitrate between parametric knowledge and in-context knowledge. Using a synthetic biographies dataset, the authors train small transformer models from scratch and systematically vary three data properties: intra-document repetition, factual inconsistency, and knowledge distribution skew. The key findings are: 1) Intra-document repetition is essential for the co-emergence of both PK and ICK capabilities. 2) A small amount of factual inconsistency in the training data induces a shift, causing the model to prefer its confident PK over conflicting ICK. 3) A skewed (Zipfian) knowledge distribution preserves the model's ability to use ICK for rare or unseen entities, preventing a total collapse to PK-only recall. Together, these properties foster a robust arbitration strategy where models trust their internal knowledge for familiar concepts but defer to the context for novel ones.

**Strengths:**

1. The research question is very important. The work addresses a critical gap by investigating how knowledge arbitration strategies are formed during pretraining.
2. The use of a synthetic dataset allows for the precise isolation and manipulation of variables (repetition, noise, skew).
3. The findings are concrete and have direct implications for data curation.

**Weaknesses:**

1. The conclusions are drawn from a small 8-layer model trained on a narrow, synthetic dataset. It is unclear if these specific dynamics, such as the degradation of in-context knowledge utilization with noise, will hold for large-scale models trained on diverse, web-scale data. The validation in Section 6 is a good first step but is too simplistic to be conclusive.
2. The study focuses exclusively on pretraining, yet the behavior of modern LLMs is heavily shaped by post-training alignment (e.g., instruction tuning, RLHF). These alignment stages could substantially alter or even overwrite the arbitration tendencies learned during pretraining, making the paper's conclusions potentially biased or incomplete for understanding deployed models.
3. The experimental setup, while clean, is limited. The conclusions about noise and skew might be sensitive to other hyperparameters or architectural choices not explored. More extensive ablation studies are needed to establish the robustness of the findings across different settings, especially for such a big title.
4. The related work section is not comprehensive and misses several critical, recent papers on knowledge conflicts and the interplay between parametric and contextual knowledge. Key omissions include:

- Understanding the interplay between parametric and contextual knowledge for large language models

- Resolving knowledge conflicts in large language models

- Adaptive chameleon or stubborn sloth: Revealing the behavior of large language models in knowledge conflicts

- ALCUNA: large language models meet new knowledge

- Evaluating the external and parametric knowledge fusion of large language models

**Questions:**

1. The study focuses on factual, entity-attribute knowledge. How might these principles apply to more abstract or procedural knowledge (e.g., mathematical or commonsense reasoning)?
2. Given that some "noise" is beneficial, what practical guidelines would you recommend for curating a pretraining corpus to achieve a desirable balance without compromising the model's overall factuality?

---

### Official Review · Reviewer_ai9i · 2025-10-31

**Soundness:** 3
**Presentation:** 3
**Contribution:** 3
**Rating:** 4
**Confidence:** 4

**Summary:**

This paper studies the training dynamics of parametric vs. in-context knowledge utilization in language models. While prior work has explored how pretrained LLMs handle conflicting knowledge, this paper is the first to systematically examine how such arbitration behaviors emerge during training under controlled conditions.

Using a synthetic biographies dataset, the authors train transformer models from scratch and manipulate corpus properties such as intra-document repetition, factual inconsistency noise, and Zipfian frequency skew, observing that:

- Models tend to rely on in-context knowledge in the early stages of training and on parametric knowledge in later stages.
- Intra-document repetition is critical for observing that phenomenon.
- Small inconsistency noise encourages preference for parametric knowledge under conflicts between context and parametric knowledge.
- If sampling of datapoints is modified towards less frequent ones, reliance on in-context knowledge can be preserved.
- The findings generalize partially to a real-world model (Pythia-6.9B) on a country–capital knowledge set.

**Strengths:**

- The work identifies an important and underexplored question: how models learn to arbitrate between stored and retrieved knowledge, rather than analyzing this post-hoc.

- The synthetic dataset allows fine-grained manipulation of repetition, inconsistency, and distributional skew—yielding interpretable insights.

- The observed phenomenon that models tend to rely on in-context knowledge in the early stages of training and on parametric knowledge in later stages is intuitively explained.

**Weaknesses:**

I can not recommend the paper for acceptance yet, unfortunately. I find the phenomenon identified in this paper genuinely interesting, but the current evidence is not fully convincing. Experiments on realistic data remain shallow, leaving uncertainty about how often this effect occurs in real-world settings. Moreover, the underlying cause of the shift between in-context and parametric knowledge remains unclear — the paper lacks theoretical grounding and mechanistic or interventional experiments that could substantiate causal claims about the observed preference transition.

## Empirical depth is insufficient.
The claim about the phase transition from ICK to PK reliance requires a stronger empirical backing in a realistic scenario.
Please, include at least one more architecture beyond Pythia for real-world validation.
In addition to that, please extend real-world evaluation beyond country–capital pairs to another domain (e.g., celebrities, historical events, animals) to ensure that the observed shift is not dataset-specific.

## Lack of theoretical grounding.
There is a limited explanation for why the model’s arbitration preference reverses during training. The intuition about “confidence growth” is plausible but purely descriptive. A more rigorous analysis could bound the model’s preference for in-context learning as a function of its confidence in predicting entity attributes. Since confidence in the correct attributes of training entities increases during training by design, if this growth correspondingly lowers the upper bound on in-context preference, then the observed phase transition would follow inevitably.

## Missing mechanistic probes.
The paper hypothesizes that induction heads drive early ICL and that confidence in parametric memory later dominates because of key-value circuits learned by the model, but there is no interventional evidence. For instance:

If one were to blur/smear keys in the attention layers (as in [1]), ICL should appear earlier. If the model had only one layer, ICL should not emerge, and $Acc_{ICKU}$ should remain flat. If you adjust the architecture to make it easier to store information
about a particular subject in a key–value format [2], parametric knowledge preference should grow faster.
These interventional experiments would strengthen causal claims about the mechanisms behind the observed shifts.

## Minor weaknesses
### Limited analysis of parameter sensitivity.

- Figure 11: The curves’ relative ordering under different noise levels remains unchanged; it would be useful to increase noise magnitude until a qualitative regime change is observed.

- Zipfian results (Figures 5, 6, 12): The trajectories for low-frequency entities resemble early-stage behavior of high-frequency ones. This suggests that longer training might eventually yield the same convergence ($Pref_{PK}$ ↑, $Pref_{ICK}$ ↓). Testing extended training (30k or 50k training steps) or extrapolating learning curves could clarify whether frequency merely delays, rather than prevents, the preference shift.

### Incomplete quantitative validation.
The authors should supplement their accuracy-based metrics with the ICL score from [1] — the loss at the 500th token minus the average loss at the 50th token in the context, averaged over examples. This would allow more standardized comparison to prior literature.

[1] Olsson, Catherine, et al. "In-context learning and induction heads." arXiv preprint arXiv:2209.11895 (2022).

[2] Zucchet, Nicolas, et al. "How do language models learn facts? Dynamics, curricula and hallucinations." arXiv preprint arXiv:2503.21676 (2025).

**Questions:**

- Why are only “birth date” and “major” attributes perturbed in the synthetic setup? How do you justify this choice?

- Can the phase shift between ICK and PK preference be predicted—for instance, by a confidence threshold or entropy gap?

---

### Official Review · Reviewer_ydio · 2025-11-10

**Soundness:** 2
**Presentation:** 2
**Contribution:** 1
**Rating:** 2
**Confidence:** 3

**Summary:**

The paper presents an analysis of how language model's capabilities to utilize parametric knowledge, i.e., knowledge learnt during training and saved in the model parameters, and in-context knowledge in a retrieval augmented generation setting given a cloze-style sentence completion task. The paper defines a synthetic task which allows for understanding when knowledge is parametric vs. in-context, and defines quantitative measures for respective knowledge utilization as well as behavior during knowledge conflicts. Using these measures, the paper studies how the knowledge utilization behaves as training progresses and how the repetition of knowledge within the training data influences the utilization capabilities.

**Strengths:**

The problem of understanding how models behave under knowledge conflicts is highly relevant for the robust and secure deployment of language models. Looking at this question through a lens of training dynamics to understand how training recipes may affect such abilities is interesting and possibly actionable (e.g. having recommendations on how to build a training dataset for specific capabilities, or schedule training).

**Weaknesses:**

- Modern RAG models are instruction-tuned to follow instructions to make use of the context. Since the authors work with pre-trained models (Pythia 6.9B) or 8-layer transformer models trained from scratch which are solely trained on the causal language modeling loss, I am doubtful whether the insights of the paper still hold for (1) instruction-tuned models that may prefer in-context knowledge over parametric knowledge (2) the widely spread larger scale models where training dynamics may differ due to scale. I understand that experiments with large-scale models are difficult to do, but it would add to a future iteration of the paper to study how their insights behave with scale (e.g. by repeating the experiments with larger or smaller models of the Pythia family). However, I believe that such an analysis would be out of scope for the rebuttal period here and I would like to stress that I do not expect such an analysis from the authors.
- Missing references (there has been more work looking into model behaviors under knowledge conflicts, which particularly corresponds to the question of knowledge conflict resolution posed in this paper), which are particularly relevant to the discussion and hypothesis in lines 317-319 :
	- Longpre et al.: Entity-based knowledge conflicts in question answering. EMNLP 2021. (This paper, to the best of my knowledge, may have coined the knowledge conflict in RAG settings).
	- Xie et al.: Adaptive chameleon or stubborn sloth: Revealing the behavior of large language models in knowledge conflicts. ICLR 2024. (esp. see the confirmation bias)
	- Kortukov et al.: Studying Large Language Model Behaviors Under Context-Memory Conflicts With Real Documents. COLM 2024. (esp. see the parametric bias)

Formatting nits (not important for the score):
- line 186: space missing before referring to Table 4

**Questions:**

There are two question blocks related to the setting and assumptions made (implicitly) in the paper:

1. Instruction-tuning:
- How do the analysis and insights of the paper hold for instruction-tuned models, where models are taught to follow instructions such as “use the following context to answer the question”?
- Do causal language modeling pretraining dynamics still influence the abilities of the model for using parametric vs. in-context knowledge, or do any learnt dynamics get overwritten by the instruction fine-tuning?
- Were all models trained solely with the causal language modeling objective or also any instruction following objectives?

2. Implicit assumption of correct parametric knowledge: In the setting of the paper, I understand that the authors assume memorized parametric knowledge to be correct and correctly related to the test task.
- What if the knowledge is not correct, as in the case of hallucinations?
- Is there a difference in the dynamics and observed behavior depending on whether the model was able to complete the test task successfully? Could the test task performance be a possible confounder in the paper's analysis?
- With the introduction of the noise setting, did the model's test task performance decrease?
- Line 53: The authors mention that we should be able to discern when to rely on parametric knowledge vs in-context knowledge solely by internal signals and not external ones. Could you please elaborate why this is desirable, given that perhaps parametric knowledge is also possible unreliable or learned in a way that’s unrelated to the question at hand?

---

### Author Response · Authors · 2025-12-03
**General Response by Authors**

We thank all reviewers for their constructive feedback. During the rebuttal period, we have made the following improvements to address the main comments:

---
## 1. Expanded Real-World Model Validation
We extended our validation beyond Pythia-6.9B to most Pythia suite (70M to 6.9B parameters). We introduce a preference gap metric (Pref_ICK - Pref_PK) showing that all model scales exhibit the same phase transition pattern: initial in-context knowledge preference followed by a shift toward parametric knowledge preference. Larger models show stronger parametric knowledge preference at convergence, consistent with prior works.

---
## 2. Post-Training Analysis
We added a new subsection examining whether arbitration strategies can be modified after pretraining:

- Real-world instruction-tuned models: Instruction-tuned models show shifts from parametric to in-context knowledge preference after instruction tuning.

- Controlled experiments: We demonstrate bidirectional modification on synthetic models: (1) noised-pretrained models shift toward in-context preference with clean post-training data, and (2) clean-pretrained models develop parametric preference with noised post-training data at varying levels (1%, 5%, 10%, 20%).

---
## 3. Improved Correlation-Based Analysis

We revised Section 3.3 to use correlation analysis between parametric knowledge confidence and conflict resolution preference. Models trained with noise show positive correlation (slope ~0.33), while models without noise show near-zero correlation (slope ~0.02). This metric is applied consistently across both Zipfian distribution analysis and post-training experiments.

---
## 4. Structural Improvements

We have substantially reorganized the paper structure to better integrate these new findings:
- Moved detailed attention mechanism analysis to the Appendix to maintain focus in the main text
- Restructured experimental sections (3.1-3.4) to provide a more coherent narrative


---
Thanks to the reviewers' valuable feedback, we have conducted extensive additional experiments and analyses during the rebuttal period. These additions—particularly the post-training analysis and expanded real-world validation—significantly strengthen our work but also require more substantial integration into the overall narrative than the current revision timeline allows.

---

### Note · Authors · 2026-01-05

**Comment:**

We would like to withdraw this submission. Thank you to the reviewers and the program committee for their time and consideration.

**Withdrawal Confirmation:**

I have read and agree with the venue's withdrawal policy on behalf of myself and my co-authors.